# Can Probiotics, Particularly *Limosilactobacillus fermentum* UCO-979C and *Lacticaseibacillus rhamnosus* UCO-25A, Be Preventive Alternatives against SARS-CoV-2?

**DOI:** 10.3390/biology12030384

**Published:** 2023-02-28

**Authors:** Héctor Valdebenito-Navarrete, Victor Fuentes-Barrera, Carlos T. Smith, Alexis Salas-Burgos, Felipe A. Zuniga, Leonardo A. Gomez, Apolinaria García-Cancino

**Affiliations:** 1Laboratory of Bacterial Pathogenicity, Department of Microbiology, Faculty of Biological Sciences, Universidad de Concepción, Concepción 4070386, Chile; 2Department of Microbiology, Faculty of Biological Sciences, Universidad de Concepción, Concepción 4070386, Chile; 3Department of Pharmacology, Faculty of Biological Sciences, Universidad de Concepción, Concepción 4070386, Chile; 4Department of Clinical Biochemistry and Immunology, Faculty of Pharmacy, Universidad de Concepción, Víctor Lamas 1290, Concepción 4030000, Chile; 5Laboratory of Molecular Immunology, Department of Microbiology, Faculty of Biological Sciences, Universidad de Concepción, Concepción 4070386, Chile

**Keywords:** probiotic, SARS-CoV-2, immunomodulation

## Abstract

**Simple Summary:**

Infection with the SARS-CoV-2 virus causes COVID-19 in humans. It has rapidly propagated worldwide, becoming a pandemic as designated by the WHO. SARS-CoV-2 is a single-stranded RNA virus belonging to the family Coronaviridae, subfamily Orthocoronaviridae, characterized by spike (S) proteins, which allow it to bind to the ACE2 receptor present on epithelial cells. Thus, this virus preferably infects tissues that highly express the ACE2 receptor, the lungs and intestine being the most affected, producing an impaired immune response or intestinal dysbiosis, respectively. Presently, treatment against COVID-19 is based on using antivirals, antibiotics to control a secondary bacterial infection, and drugs that attack the symptomatology associated with the infection, alone or combined. Furthermore, several vaccines have been approved that show variable effectiveness, through a diverse number of doses, in confronting the emergence of new SARS-CoV-2 variants; however, this effectivity is affected by the low numbers of vaccinated individuals in developing countries, rates that could increase due to a lack of reinforcements. Considering the present situation, new alternatives with which to treat this problem have been proposed. These alternatives include probiotics, particularly immunobiotics, which can beneficially stimulate intestinal immunity and pulmonary immunity through the intestine–lung axis.

**Abstract:**

COVID-19, an infection produced by the SARS-CoV-2 virus in humans, has rapidly spread to become a high-mortality pandemic. SARS-CoV-2 is a single-stranded RNA virus characterized by infecting epithelial cells of the intestine and lungs, binding to the ACE2 receptor present on epithelial cells. COVID-19 treatment is based on antivirals and antibiotics against symptomatology in addition to a successful preventive strategy based on vaccination. At this point, several variants of the virus have emerged, altering the effectiveness of treatments and thereby attracting attention to several alternative therapies, including immunobiotics, to cope with the problem. This review, based on articles, patents, and an in silico analysis, aims to address our present knowledge of the COVID-19 disease, its symptomatology, and the possible beneficial effects for patients if probiotics with the characteristics of immunobiotics are used to confront this disease. Moreover, two probiotic strains, *L. fermentum* UCO-979C and *L. rhamnosus* UCO-25A, with different effects demonstrated at our laboratory, are emphasized. The point of view of this review highlights the possible benefits of probiotics, particularly those associated with immunomodulation as well as the production of secondary metabolites, and their potential targets during SARS-CoV-2 infection.

## 1. Introduction

### 1.1. SARS-CoV-2 and Probiotics

The viral infection caused by SARS-CoV-2 rapidly spread throughout the world, prompting the WHO to classify it as a pandemic. SARS-CoV-2 belongs to the *Orthocoronaviridae* subfamily, a member of the *Coronaviridae* family. SARS-CoV-2 is a single-stranded RNA virus whose genome corresponds to approximately 27–32 kb. It uses the spike protein to infect lung and intestinal epithelial cells. These cells possess a membrane receptor protein, angiotensin-converting enzyme 2 (ACE2), used as an entry point by some coronaviruses. The viral RNA is released into the cytoplasm of host cells, where the required viral proteins, either structural ones or those for the replication of its genetic material, are transcribed [1]. Since ACE2 is highly expressed by the lungs and the intestine, they are the most affected organs. SARS-CoV-2 infection causes dysbiosis and an uncontrolled immune response, which may even cause the death of an individual [2].

Currently, infection by SARS-CoV-2 is treated using hydroxychloroquine, tocilizumab, antivirals, antibiotics, and treatments to handle the symptomatology associated with the infection, either alone or in combinations, sometimes without success [3]. This fact has encouraged a number of companies and researchers to collaboratively develop vaccines that are administered in different doses and that have shown different percentages of effectiveness against new variants of the virus. Among the variants, Alpha (known as 501Y.V1 according to the GISAID nomenclature or variant B.1.1.7 according to the Pango nomenclature), Beta (501Y.V2 or B.1.351), Gamma (501Y. V3 or P1), and Delta (G/478K.V1 or B.1.617.2) can be mentioned [3].

Taking into consideration the ineffectiveness of anti-COVID-19 drugs and differences in the doses as well as the effectiveness of the vaccines, resulting from the advent of new variants, new therapeutic alternatives have been sought. One of these proposed therapeutic alternatives is the administration of probiotics [4]; even more specifically, immunobiotics. Immunobiotics are probiotic strains capable of beneficially regulating the mucosal immune system [5]. Considering the fact that the intestinal microbiota may beneficially modulate intestinal immunity, as well as pulmonary immunity by means of the intestine–lung axis [6], there emerges the possibility of taking advantage of the immunomodulating properties of immunobiotic bacteria. Immunobiotic bacteria, when colonizing the gastrointestinal tract of an individual, could induce a beneficial effect on the immune system of a host at both the gastrointestinal and respiratory systems. This result is due to the fact that the local immunological regulation by the specific microbiota of the gut has a long-range immunological impact that reaches the lung [7]. Therefore, probiotics may provide a wide scope of benefits to consumers, including the prevention and treatment of infections of the upper respiratory tract [8].

### 1.2. SARS-CoV-2: A High-Prevalence Threat to Global Public Health

The emergence of new viruses capable of causing human respiratory diseases has turned these pathologies into global challenges. Respiratory viral infections are among the most prevalent ones, contributing significantly to the morbidity and mortality rates in all age groups. Furthermore, they can rapidly evolve and cross species barriers into other host populations. Moreover, they are associated with severe clinical diseases and mortality [9]. Since there are no means with which to control emerging infectious diseases, they are problematic in terms of being treated and efficiently prevented. Among these emergent respiratory pathologies, it is possible to include severe acute respiratory syndrome (SARS), which emerged in 2002, infected 8098 persons in 26 countries, and caused 774 deaths, its causative agent being the SARS-CoV-1 virus. Later on, in 2012, coronavirus reappeared in the Arabian Peninsula in the Middle East (MERS-CoV virus), and most recently, in 2019, in China (SARS-CoV-2 virus). SARS-CoV-2 is the causative agent of COVID-19 [10], becoming a worldwide public health emergency [11].

By October 2022, SARS-CoV-2 had caused nearly 61,577 deaths as a consequence of approximately 4,722,184 cases in Chile, while Cuba suffered a total of 8530 deaths as the result of 1,111,290 cases, Argentina recorded 129,991 deaths caused by 9,718,875 cases, and Uruguay reported a total of 7518 deaths due to 990,560 cases. These countries are reaching the higher percentages of complete immunization schemes in Latin America, led by Chile, with 91.46% of its total population being vaccinated. At the worldwide level, it is estimated that the numbers of cases and deaths reach figures of approximately 629,000,000 and 6,590,000, respectively [12,13].

By the end of February 2021, 1755 cases requiring treatment at an intensive care unit were reported in Chile, 1513 of which required invasive mechanical ventilation, the highest figures recorded in the country during the whole pandemic [13].

Considering its known epidemiological background, SARS-CoV-2 is an excellent model with which to develop strategies to control emerging severe viral infections, centering efforts to prevent disease. Among the characteristics of this virus, we will discuss possible targets identified to develop these strategies.

### 1.3. SARS-CoV-2 Infection Generates an Uncontrolled Immune Response

SARS-CoV-2 is a pneumotropic virus transmitted from person to person through respiratory secretions, including droplets produced while coughing, sneezing, or even speaking. Contact with contaminated surfaces or fomites also contributes to its transmission [14,15]. In closed environments with inadequate ventilation SARS-CoV-2 is a highly infectious pathogen, being in potentially infectious aerosols for hours and possessing the ability to travel for tens of meters before landing on surfaces where the virus can survive for days [16,17].

SARS-CoV-2 infection occurs due to the binding of its spike protein with the angiotensin-converting enzyme 2 (ACE2) receptor present on epithelial cells. This infection can be divided into three phases: (1) asymptomatic—the virus being detectable or not; (2) symptomatic—non-severe, presence of detectable virus, including infection, fever, dry cough, myalgia, fatigue, and breathing difficulties, occasionally diarrhea, nausea, and vomiting [18], patients showing gastrointestinal symptoms being the ones with the worse prognosis [19]; and (3) symptomatic—severe respiratory symptoms due to a high viral load [20] cause a storm of cytokines in the immune system. The infection is characterized by high serum levels of IL-1β, IL-2, IL-7, IL-8, IL-9, IL-10, IL-17, G-CSF, GM-CSF, IFN-γ, TNFα, IP10, MCP1, MIP1A, and MIP1B, but, in particular, it influences a sustained increase in IL-6 and IL-1β [21,22,23], which may result in the death of a patient. This immune response generates uncertainty with regard to the evolution of a patient’s diagnosis; for this reason, the Dublin–Boston scoring parameter was developed. It allows us to predict the probability that a patient will require hospitalization, intensive care, or mechanical ventilation, for which the blood IL-6 and IL-10 quotients during infection are calculated [24].

On the other hand, the humoral immune response is mediated by antibodies, which are detectable approximately between four and six days after RT-PCR, confirming an infection. Anti-SARS-CoV-2 antibodies show neutralizing capacity, can contribute to eliminating the virus, and, subsequently, to preventing the disease. Nevertheless, it is yet uncertain if they provide long-lasting immunity and protection against reinfection [25], because, in mild COVID-19 patients, the IgG antibodies’ response against the virus decreases rapidly (2 to 4 months), suggesting that, in these patients, this may be a short time response [26]. This short time response may be the consequence of the loss of Bcl-6-expressing T follicular helper cells and germinal centers in patients suffering mild COVID-19 infection [27].

### 1.4. SARS-CoV-2 Infection Causes Intestinal Dysbiosis Associated with a Loss in Intestinal Mucosa

The main symptoms associated with the SARS-CoV-2 infection include fever, dry cough, myalgia, fatigue, respiratory distress, and, less frequently, diarrhea, nausea, and vomiting [19]. Besides elderly patients, patients with underlying pathologies, such as chronic obstructive pulmonary disease, diabetes, or cardiovascular diseases, may develop clinical presentations that are much more severe. These include acute respiratory distress syndrome and septic shock, which might lead to a patient’s death [20]. In the intestine an infection causes nausea, diarrhea, and vomiting; moreover, recent studies suggest that SARS-CoV-2 alters the intestinal microbiota [21]. This alteration is possibly due to the reduced availability of ACE2 during an infection by this virus, which could be sufficient to alter the composition of the intestinal microbiota [22]. Some evidence also supports the possibility that SARS-CoV-2 may induce alterations in the blood barrier of the intestine, promoting the abnormal absorption of microbial molecules or the dissemination of bacteria. This stimulates the systemic inflammatory response, which, in turn, contributes to multiorgan dysfunction, septic shock, or gastrointestinal and fecal dysbiosis [28].

A consequence of dysbiosis is the colonization of the intestine by opportunistic microorganisms, such as members of the genera *Parabacteroides*, *Faecalibacterium*, and *Clostridium*, to the detriment of beneficial commensal genera, such as those belonging to the phylum Firmicutes. Under normal circumstances, ACE2 participates in the renin–angiotensin system (RAS), hydrolyzing Ang II into Ang (1-7) [29], which is involved in the inhibition of glucose transport to the intestine and in the production as well as release of insulin in the presence of high glucose levels [30]. During SARS-CoV-2 dysbiosis, due to the blocking of ACE2 by the virus, there occurs a decrease in the concentration of Ang (1-7), which reduces the availability of insulin and increases intestinal glucose, contributing to leaky gut syndrome and causing an increase in intestinal glucose. Figure 1 shows a diagram of the intestinal dysbiosis process.

### 1.5. Current Strategies to Deal with COVID-19

#### 1.5.1. Vaccines, the Only Preventive Therapy to Deal with COVID-19

Up to 15 February 2021, more than 200 vaccines were in the process of being developed and 50 of them were being subjected to clinical phases; only 11 had progressed to phase 3 and become authorized to be used in the population [31]. Reports from 3 March 2021 indicate that only twelve have been approved by at least one country to immunize their population. In Chile, vaccination against SARS-CoV-2 started on 24 December 2020, and, so far, in this country only five vaccines have received emergency approval: BNT162b2 (developed by Pfizer, Pearl River, NY, USA, EE.UU. and BioNTech, Mainz, Germany), CoronaVac (produced by Sinovac Biotech, Pekin, China), ChAdOx1-S (produced by AstraZeneca, Cambridge, UK and Oxford, Oxford, UK) [32], CanSino (produced by CanSino Biologics Inc., Tianjin, China), and mRNA-1273 (produced by Moderna, Cambridge, MA, USA, EE.UU.). Vaccines are the only prevention strategy being developed against COVID-19. It is noteworthy that these vaccines possess certain negative aspects; for example, a few individuals will not be able to receive the vaccine, such as those with severe allergies to any of the vaccine components [33,34,35]. The probability of the occurrence of mutations of the virus does exist, and vaccines may lose effectiveness against these new variants [36]. Finally, it is still unknown as to how long-lasting immunity is against the virus after vaccination, nor when the required fraction of the population will be vaccinated with adequate doses to have the virus and its variants under control [37].

#### 1.5.2. Current Treatments to Control SARS-CoV-2 Infection

The redeployment of drugs has been proposed; that is to say, to analyze the possibility of using drugs already in the market as antivirals or for other purposes, to validate their efficacy against SARS-CoV-2, and, if appropriate, to accelerate their application against COVID-19 [38]. Among these drugs, we can find remdesivir (GS-5734; Gilead Sciences Inc., Foster City, CA, USA) and favipiravir (Avigan, Toyama Chemical. Co. Ltd., Tokyo, Japan), which, as part of their primary functions, inhibit the replication of the virus and control the storm of cytokines caused by the infection [39,40].

#### 1.5.3. Co-Adjuvant Treatments against COVID-19

The use of co-adjuvants, such as high doses of vitamin D [41], intravenous vitamin C [42], zinc supplements [43], or medicinal plants and their active components, has been proposed as a supplementary treatment [44]. They are under investigation, but none have been accepted as being a definitive treatment. Reasons to preclude their use so far include only a short period of study, insufficient evidence to correctly justify their action mechanisms against the virus, and a lack of knowledge regarding possible adverse effects that they could cause in the population, mainly in the risk groups.

Therefore, up to the present, there are no highly efficient therapies against SARS-CoV-2 infection. In general, prevention is the most efficient therapy against viral infection. Thus, probiotics emerge as appropriate candidates to control this infection [45]. In this review we will highlight the importance and relevance of probiotics as appropriate candidates based on the evidence presented below.

## 2. Materials and Methods

A search was conducted in the literature by using the keywords probiotic; SARS-CoV-2; and immunomodulator. The search covered the time span of 2015 to 2022, across different search engines and databases. English and Spanish were the languages selected. We expected to find results for SARS-CoV-2 from the year 2020 on, but we required information on the other keywords from before 2020.

Information published in the databases about the immunobiotic strains *Limosilactobacillus fermentum* UCO-979C and *Lacticaseibacillus rhamnosus* UCO-25A was gathered.

A search on intellectual property and patents was performed based on the keywords probiotic; SARS-CoV-2; and immunomodulator, using PATENTSCOPE of the WIPO database (URL: https://patentscope.wipo.int/search/es/search.jsf, (accessed on 10 October 2022)).

Based on the information presented in this review, we performed an in silico analysis, searching for the presence of genes related to metabolites with possible anti-COVID-19 functions within the genomes of both probiotic strains—*Limosilactobacillus fermentum* UCO-979C and *Lacticaseibacillus rhamnosus* UCO-25A. For this, we annotated a draft of the genomes of both strains with Prokka [46] and generated output files that allowed for a posterior analysis of the genomes. Once annotated, we used the eggNOG-mapper tool [47] to obtain a metagenomic catalog of possible genes related to enzymes or metabolites with possible effects against SARS-CoV-2.

All figures were produced using the free bioicons library by Simon Duerr (https://bioicons.com, (accessed on 10 October 2022)).

## 3. Results

### 3.1. Probiotics, an Alternative

Antibiotics and palliative treatments aim to treat the diseases caused by respiratory infections instead of preventing them. Besides favoring the emergence of resistance in microorganisms, these treatments have other drawbacks. Therefore, other alternatives, such as probiotics, appear as exciting possibilities. Under the Food and Agriculture Organization of the United Nations (FAO) and the World Health Organization (WHO), probiotics are defined as “live microorganisms which, when administered in adequate amounts confer a health benefit on the host” [48]. More specifically, immunobiotics are probiotics capable of beneficially regulating the mucosal immune system [49]. Probiotics have become subjects of large amounts of interest after meta-analysis studies have demonstrated that consuming them can prevent gastrointestinal problems, such as antibiotic-associated diarrhea and bacterial as well as viral infections, such as sepsis and respiratory tract infections [50,51]. Figure 2 shows the mechanism of action of probiotics.

### 3.2. Effect of Probiotics on the Intestine–Lung Axis

Since the intestinal microbiota can beneficially modify intestinal immunity as well as lung immunity through the intestine–lung axis [6], the immunomodulating properties of immunobiotic bacteria colonizing the intestinal tract will benefit the gastrointestinal as well as respiratory tract immunity of a host. This occurs because the local specific microbiota have a long-range immunological impact, going from the intestine to the lungs [52]. This axis occurs thanks to the mesenteric lymphatic system, the path between these two organs, as shown in Figure 3 [53], allowing intact bacteria, their fragments, or their metabolites to cross the intestinal barrier to reach the systemic circulation, and thus have an influence on the immune response of the lungs [7]. In the case of respiratory tract infections, it is possible to have an influence by means of the intestine–lung axis because probiotics stimulate the immune response at the local and systemic levels. Hence, the stimulation at the intestinal level will allow the intestinal microbiota to modulate the immune system during a respiratory tract infection [54]. That is to say, certain probiotic strains prevent viral infections of the respiratory and gastrointestinal tracts, avoiding infections associated with these mucosae [5,55,56,57,58,59].

### 3.3. Some Probiotics Might Be a Complement to Current Therapies

The effect of immunomodulating probiotics on controlling infectious diseases of the respiratory tract has been demonstrated [60]. Table 1 describes the therapeutic actions of probiotics against SARS-CoV-2 as oral bacteriotherapy complementary treatments for COVID-19 [61]. Nevertheless, to the best of our knowledge, no studies have been reported as describing immunomodulatory or preventive effects of probiotic strains against infections caused by SARS-CoV-2.

### 3.4. Probiotics May Antagonize SARS-CoV-2 Infection

Considering the information analyzed above, it is known that the intestinal microbiota can beneficially alter intestinal immunity and pulmonary immunity by means of the intestine–lung axis [54]. Thus, the use of immunobiotics to treat COVID-19 has very much become relevant at a worldwide level [5]. Figure 4 shows the possible effect of probiotics against SARS-CoV-2.

Oral bacteriotherapy has resulted in positive and encouraging results; for example, in patients hospitalized at the Department of Infectious Diseases, Umberto I Polyclinic, University “Sapienza”, Rome, Italy. In this study, 28 patients received Sivomixx as a supplement to their treatment. Sivomixx is a product containing *Streptococcus thermophilus* DSM 32345, *Lactobacillus acidophilus* DSM 32241, *Lactobacillus helveticus* DSM 32242, *Lactobacillus paracasei* DSM 32243, *Lactobacillus plantarum* DSM 32244, *Lactobacillus brevis* DSM 27961, *Bacillus lactis* DSM 32246, and *Bacillus lactis* DSM 32247. Additionally, 42 patients received antibiotics, hydroxychloroquine, or tocilizumab treatment without oral bacteriotherapy. Patients who received Sivomixx showed better survival and a reduced risk than patients that did not receive the probiotic treatment. Additionally, the estimated risk of developing respiratory insufficiency while having COVID-19 was reduced by more than eight times in the group receiving oral bacteriotherapy when compared to non-treated patients [61].

### 3.5. Possible Targets and Metabolites of Probiotics Related to SARS-CoV-2 Infection

Based on the above, the possibility of using probiotics to confront COVID-19 has become attractive to researchers. Therefore, several studies have focused on the search for possible therapeutic targets and metabolites produced by probiotics whose function could be of interest against SARS-CoV-2.

#### 3.5.1. Possible Targets

Besides ACE2, which has been considered to be the main mediator for SARS-CoV-2 infection, allowing the entry of the virus into cells, there are a number of therapeutic targets that have become relevant after recent studies on infection by this virus. These targets include different proteins related to the mechanisms of infection of the virus and their entry into cells, such as the receptor binding domain (RBD) and the main protease 3CLpro, as well as other proteins related to replication, such as non-structural proteins (NSPs).

##### RBD

The spike glycoprotein of coronaviruses is the main glycoprotein involved in the entry of the virus into cells. It includes two subunits, named S1 and S2. S1 contains the RBD, which is the domain involved in the entry of the virus into cells [74]. The entry occurs by the binding of RBD and ACE2 [75], the main cell receptor used during infection by a coronavirus, favoring the membrane fusion by means of S2. Therefore, RBD–ACE2 binding is critical for infection [76]. In silico studies on the molecular dynamics between metabolites and proteins related to SARS-CoV-2 infection have demonstrated the potential use of plantaricin, a metabolite produced by bacteria belonging to the *L. plantarum* species, to inhibit SARS-CoV-2 infection due to its ability to bind to the viral RBD, ACE2, and RNA-dependent RNA polymerase (RdRp) [77].

##### 3CLpro

The coronavirus genome includes the sequence of a main protease, denominated 3CLpro or SARS-CoV 3CLpro, which is essential for the life cycle of this virus [78]. This protease cleaves the polyproteins of the virus to generate the diverse non-structural proteins responsible for the replication of the virus [79], viral dissemination, and the inhibition of the interferon (INF) response [80]. Combined, these characteristics turn 3CLpro into a potential target to fight against SARS-CoV-2. Furthermore, studies have reported that some compounds are capable of inhibiting this protease. This is the case for compounds containing boronic acid (aryl boric acid), which are capable of binding to the serine cluster near to the active site of this protease, inhibiting its activity [78], for lactoferrin (antimicrobial peptides), and for lactococcin (bacteriocins), which, based on interaction studies, are capable of inhibiting this protease [81].

##### Non-Structural Proteins (NSPs)

The genome of SARS-CoV-2 codifies several polyproteins that, by means of the activity of the protease 3CLpro, generate 16 types of NSPs (named NSP1 to NSP16). These NSPs are mainly related to the viral replication and transcription mechanisms [82]. Two of these NSPs stand out as interesting targets with which to battle against SARS-CoV-2 infection. They are NSP13 helicase and NSP12 viral RdRp.

NSP13 is a helicase, highly conserved within the coronavirus family, which has been demonstrated to be essential for the replication of the virus [82]. Structural, crystallographic, and mechanistic studies have identified two pockets where ATP and RNA bind [83]. These pockets and ATPase activity have been the target of multiple studies, resulting in several drugs and metabolites having the potential to bind to them, reducing the activity of NSP13 helicase [83,84]. Moreover, studies by Sui et al. [85] showed that NSP13 is capable of inhibiting the production of type I IFN, allowing the virus to evade the innate immune response, shedding lighter on the importance of this protein in the pathogenesis of the virus and reinforcing its potential as a therapeutic target.

Concerning NSP12, a RdRp that plays a key role in the cycle of the SARS-CoV-2 virus, it is the main protein in the coronavirus replication complex and has NSP7 as well as NSP8 as cofactors [86]. Nevertheless, it has been shown that SNP12 by itself can exhibit basal activity [87]. Thus, NSP12 is a potential target in anti-SARS-CoV-2 treatment. Hence, some studies have focused on the search for drugs capable of inhibiting its polymerase activity [88], and the initial approaches may indicate that an effective blockade of NSP12 could inhibit the replication of the virus.

#### 3.5.2. Possible Metabolites of Probiotics with SARS-CoV-2 Inhibitory Activity

Bioinformatics analyses have found several metabolites produced by probiotics that may prevent the infection produced by SARS-CoV-2. Among them, we can highlight peptide-type metabolites and short-chain fatty acids [89].

##### Bacteriocins from Lactic Acid Bacteria

In silico studies on *L. plantarum* antiviral metabolites obtained from PubChem, based on binding energy and molecular docking, performed by Anwar et al. [77], predicted three types of plantaricin produced by this probiotic bacterium as showing significant interaction with the RBD, a crucial segment of the spike for the binding of the spike protein with the ACE2 receptor, as well as directly with ACE2 and RdRp. In addition, Nguyen et al. [90] demonstrated, in a case study, the potential of plantaricin E and F as efficient blockers of ATP and ssRNA binding to NSP13 helicase. Considering the different studies and evidence supporting the role of *L. plantarum* as a potential probiotic bacterium with which to treat COVID-19, its therapeutic capacity as a potential adjuvant was evaluated in in vitro and in silico studies. In this context, an extract of *L. plantarum* Probio-88 was shown to be able to significantly inhibit the replication of the virus and to reduce indicators of inflammation as well as reactive oxygen species (ROS) [91]. Moreover, molecular docking showed that the antiviral activity was due to plantaricin E and F [91].

In addition, various other bacteriocins produced by lactic acid bacteria have been studied, in silico, to evaluate their possible performances as future treatments focused against SARS-CoV-2 and its variants. Thus, through docking and molecular dynamics, it was observed that pediocin PA-1, salivaricin B, and salivaricin P were promising candidates against SARS-CoV-2 and its β variant, being able to prevent the entry of the SARS-CoV-2 virus into host cells. [92].

##### Microbial Peptides

Ever since the successful results obtained with peptide hormones in treating diseases, such as in the case of insulin, and considering their lower costs as well as easier production, the market of peptide drugs and vaccines has been increasing, with more than 150 peptides being under development [93]. In this context, several studies have focused on determining the efficacy of a diversity of peptides, both natural and synthetic, to treat COVID-19 [94,95,96]. Results with respect to either the inhibition of some essential SARS-CoV-2 proteins, such as the main protease, or to the antagonism against the spike–ACE2 interaction, have been shown to be promising [95,96]. In fact, presently there is a database available of antimicrobial peptides with inhibitory effects against coronaviruses [97]. Thus, the information available makes the search for alternatives to peptide drugs with which to treat COVID-19 an appealing and strongly supported option. Probiotics, because of the peptides that they synthetize, play an essential role within this due to the fact that they are a natural alternative, avoiding the use of drugs. In silico studies have shown the potential effects of peptides derived from probiotics, such as subtilisin (*Bacillus amyloliquefaciens*), which interacts with the RBD, preventing its binding to the ACE2 receptor, and curvacin A (*Lactobacillus curvatus*), sakacin P (*Lactobacillus sakei*), and lactococcin Gb (*Lactococcus lactis*), which are able to form complexes with ACE2, preventing virus entry. All of the four peptides mentioned above act as competitive inhibitors against the interaction of SARS-CoV-2 with the ACE2 receptor [98].

##### Short-Chain Fatty Acids (SCFA)

The short-chain fatty acids (SCFA) are among the most studied metabolites produced by bacteria, possessing beneficial properties. Among them, butyrate, acetate, and propionate stand out. These types of compounds are produced by diverse types of bacteria of the microbiota present in the digestive tract [99], whose study, considering their beneficial effects, has become a subject of interest. These studies have demonstrated that butyrate is capable of maintaining a stable epithelial barrier, regulates inflammation, and acts as an immunomodulator [100]. Moreover, evidence has been found that, as part of its functions, it is involved in the response against respiratory diseases and strengthens the immune response [101,102]. Studies conducted on mice have demonstrated that, at the pulmonary level, butyrate can restore IL-10 levels as well as reduce infections [101] and immunopathology in cases of influenza [102]. Other studies have demonstrated that butyrate inhibits histone deacetylase, regulating gene expression, which becomes relevant because during SARS-CoV-2 infection there occurs a deregulation of ACE2 expression, which allows the virus to enter into the cells of the host. This deregulation has been, in turn, correlated with a decrease in butyrate-producing bacteria, which may explain the increased expression of ACE2 during SARS-CoV-2 infection [103].

### 3.6. Intellectual Property Associated with Probiotics and COVID-19

A search for patents referring to the use of bacteria to counteract COVID 19 was performed at the international level, using PATENTSCOPE of the WIPO database. The following patents were found:

WO2020143892: French patent requested by Lachlak, Nassira and Bensebti, Ishaq. It refers to the use of the colostrum of pregnant cows infected with SARS-CoV-2, which contains beneficial bacteria. This protected invention does not make reference to a specific mechanism of the bacteria of the colostrum to control SARS-CoV-2 infection, leaving this labor to the antibodies present in it. Nevertheless, the concentrations and types of bacteria may vary from one colostrum to another; thus, the replicability of its use is low.

RU2735723C1: Russian patent requested by the National Medical Research Center for Rehabilitation and Balneology. It protects the invention of a nutritional complex containing probiotic bacteria and prebiotics. The use of this complex is for the treatment of persons that already show symptoms of the disease.

WO2021074706: Colombian patent requested by ALSEC ALIMENTOS SECOS S.A.S. This patent protects the invention of a dietary supplement for foods and for the pharmacologic industries to prevent and treat diseases associated with immunodeficiency and viral infections caused by viruses such as SARS-CoV-2.

### 3.7. The Use of Probiotics Could Reduce the Economic Costs Associated with the Control of Respiratory Diseases

The efficacy of probiotics has been demonstrated when they are administered to healthy individuals to reduce the incidence and duration of respiratory infectious diseases [104,105,106]. It is estimated that the generalized use of probiotics could have reduced costs by USD 4,600,000 for the population of that country in 2017–2018, due to avoiding days of acute infections of the respiratory tract. If the prescription of antibiotics and the loss in productivity are also included in the calculations, the amount saved increases [107].

It is estimated that, in Chile, the use of probiotic-supplemented foods could contribute to saving a total annual cost of over USD 350 million, considering both public and private healthcare services, if the use of probiotics is implemented as a preventive measure [108,109]. It would also release beds at healthcare services for those patients suffering from other types of pathologies and prevent the collapse of this type of facility [110].

### 3.8. Limosilactobacillus Fermentum UCO-979C and Lacticaseibacillus Rhamnosus UCO-25A Strains Exhibit Immunological Characteristics That Are of Interest Due to Their Possible Use against SARS-CoV-2

(1) Isolation and characterization of probiotic strains. The strains *L. fermentum* UCO-979C and *L. rhamnosus* UCO-25A were isolated, 15 years ago, at our Laboratory of Bacterial Pathogenicity (Department of Microbiology, Faculty of Biological Sciences, University of Concepción, Concepcion, Chile) from human gastric biopsies provided by the Service of Gastroenterology of the Dr. Guillermo Grant Benavente Hospital (Concepcion, Chile). Both strains are capable of tolerating and surviving at a pH of 3, [111,112,113], in addition to being able to tolerate 1.5% or 2.0% bile salts for up to 24 h and possessing high antibiotic susceptibility profiles [111,113]. These results indicate their capacity to adapt, survive, and extensively colonize the gastrointestinal tract without the risk of transferring antibiotic resistance genes to the bacteria of the microbiota of this anatomical region [112,113,114].

(2) Immunobiotic properties. The capacity of both strains to modulate the immune response as the result of an infection by *Helicobacter pylori* has been evaluated in AGS cells (human gastric adenocarcinoma cell line) and THP-1 cells (human monocytic cell line). Both cell lines were demonstrated to increase the IL-10 and IFN-γ anti-inflammatory response as well as reduce the TNF-α IL-8 and IL-1b proinflammatory response, being the preventive model that provided the best results [115,116,117]. This is an important issue because the reduced production of inflammatory factors, particularly IL-8 and TNF-α, is associated with a less serious infection and ensuing sequels [115,117]. Table 2 summarizes the effects of strains *L. fermentum* UCO-979C and *L. rhamnosus* UCO-25A on the immune response.

(3) Innocuity and preventive capacity to confront *H. pylori* infection in an animal model. Both strains were demonstrated to be innocuous in a murine model [115,117]. *L. fermentum* UCO-979C was shown to stimulate the activation of peritoneal macrophages and T CD4^+^ cells, to increase IgA levels, to modulate intestinal and serum cytokines, and to reduce the number of immature B cells in Peyers’s patches [115,116,117,118,119]. On the other hand, *L. rhamnosus* UCO-25A increased the activity of peritoneal macrophages, IgA levels, the activity of dendritic cells, and B as well as T cells, and it modulated the intestinal and systemic levels of cytokines [119]. Thus, both strains could beneficially modulate the immunity of individuals.

(4) Technological parameters of the strains. It has been shown that both probiotic strains are very resistant when subjected to scaling processes, and that they maintain their probiotic properties after the different stages of processes. Both probiotic strains are easy to handle at a semi-industrial level. *L. fermentum* UCO-979C has been scaled-up from a laboratory to industrial level, while *L. rhamnosus* UCO-25A has been scaled-up at a semi-industrial level.

(5) Assays in humans. A clinical assay was performed with 121 participants, who received a *L. fermentum* UCO-979C-supplemented gelatin, supplied five times a week for 12 weeks, and showed a preventive effect against *H. pylori* infection [120]. In another assay, 20 participants were subjected to a nutritional intervention with *L. rhamnosus* UCO-25A in an oat shake, demonstrating that this strain does not alter the organoleptic properties of the shake and that it was well-received by the participants. These results suggest that both strains can be incorporated into different matrices for human consumption.

Considering the information discussed above, we can suggest that the use of these two immunobiotic strains is capable of modulating, in a positive way, the immune system of a host in a prophylactic or therapeutic manner when confronting SARS-CoV-2 infection. For this reason, an in silico study was carried out that can provide us with possible avenues of study to be subsequently tested at the in vitro and in vivo levels.

**Table 2 biology-12-00384-t002:** Comparison of the effect of SARS-CoV-2, based on previous studies on *Limosilactobacillus fermentum* UCO-979C and *Lacticaseibacillus rhamnosus* UCO-25A, on the immune response and the intestinal microbiota.

Factor		SARS-CoV-2	*L. fermentum* UCO-979C	*L. rhamnosus* UCO-25A	References
Immune response	Cytokine storm	Increased serum levels of IL-2, IL-7, IL-8, IL-9, IL-10, IL-17, G-CSF, GM-CSF, IFN-γ, TNFα, IP10, MCP1, MIP1A, and MIP1B, but particularly increases IL-6 and IL-1 levels	It modulates intestinal and systemic cytokine levels	It modulates intestinal and systemic cytokine levels	[115,116,117,119]
	IL-6	Deregulates IL-6 levels	Reduces the proinflammatory response, decreasing IL6 It possesses mixed stimulating/anti-inflammatory effects at the intestinal level, increasing IL-6	Reduces the proinflammatory response, increasing gastrointestinal IL-6	[115,116,117,119]
	IL-10	Deregulates IL-10 levels	Reduces inflammation in the affected gastric tissue, increasing IL-10	Shows anti-inflammatory activity, increasing IL-10	[115,116,117,119]
	Macrophages	Macrophage activation syndrome (MAS)	Macrophages (THP-1 cell line) show anti-inflammatory activity, reducing TNF-α and increasing IL-10 as well as IFN-γ Stimulates the activation of peritoneal macrophages	Macrophages (THP-1 cell line) show anti-inflammatory activity, reducing TNF-α as well as IL-8 and increasing IL-10 as well as IFN-γ Increases the activity of peritoneal macrophages	[115,116,117,118,119]
	T cells	Causes inadequate behavior of IFN and MHC expression pathways in macrophages, NK cells, and dendritic cells, delaying the activation as well as response of T lymphocytes and allowing increased viral replication	Stimulates the activation of T CD4+ cells	Increases the activity of T cells	[115,116,117,118,119]
	B cells	The alterations to the T response and its delayed activation, caused by viral interference on the innate response, causes an inadequate follicular B response	Reduces the number of immature B cells in Peyer’s patches	Increases the activity of B cells	[114,115,116,117,118,119]
Intestinal microbiota	Intestinal dysbiosis	Dysbiosis of the microbiota	In an animal model (Mongolian gerbils), these probiotic strains were able to colonize as well as survive in the gastrointestinal tract and maintain their viability for a long period (10 to 15 days), favoring their probiotic capacity Capable of reducing and protecting from intestinal inflammation, as well as inhibiting the adhesion and invasion of pathogens (H. pylori) Help to restore the intestinal microbiota and reduce severe gastrointestinal symptoms		[114,115,116,117,118,119]
		It can infect the gastrointestinal tract and actively replicate in it			
		In patients infected by SARS-CoV-2, 5 to 10% develop intestinal symptoms, such as diarrhea, vomiting, and abdominal pain			
	Intestinal immune response	Large release of proinflammatory cytokines and recruitment of neutrophils, macrophages, and other cell types which contribute to uncontrolled systemic inflammation and a cytokine storm Increased IL-33 and IL-8 levels in fecal samples of COVID-19 patients due to intestinal involvement in addition to a decrease in IL-1β, TNF-α, and IL-6 cytokines	At the gastrointestinal level (AGS cell line), it reduces the proinflammatory response (reduces TNF-α, IL1-β, IL-6, and IL-8)	At the gastrointestinal level (AGS cell line), it reduces the proinflammatory response (reduces TNF-α, IL1-β, and IL-8) and increases IL-6	[114,115,116,117,119]
			At the intestinal level (swine intestinal cells and murine model), mixed stimulating/ anti-inflammatory activity Reduces the expression of inflammatory factors, such as CXCL8, CXCL9, CXCL10, CXCL11, C1S, and C3, and it increases the expression of IL-6, CCL8, C1R, and CFB	In the murine model, it increases the activity of peritoneal macrophages and intestinal IgA levels	[114,115,116,117,118,119]
			At the gastric level in a murine model, it decreases the inflammation of the affected tissue, in addition to an increase in IFN-γ and IL-10 as well as a decrease in serum TNF-α and IL-8		

### 3.9. In Silico Metagenomic Analysis of Metabolites Present in Immunobiotic Strains Limosilactobacillus Fermentum UCO-979C and Lacticaseibacillus Rhamnosus UCO-25A with a Possible Effect on COVID-19 Treatment

Based on unpublished data from our work team, which helps to understand the properties of probiotic strains that can act against the pathogenicity of SARS-CoV-2, several genes of interest are highlighted; Table 3 summarizes these genes. As part of the genes of interest of the strain *L. fermentum* UCO-979C, genes related to the production of butyrate, bacteriocins, and of secondary bile acids were found. With respect to *L. rhamnosus* (the species to which the strain UCO-25A belongs), genes related to the production of short-chain fatty acids, such as butyrate, bacteriocins, and to the transport of bile acids were found.

Regarding butyrate, *L. fermentum* UCO-979C possesses genes related to the synthesis of polyhydroxybutyrate (e-value: 2.18 × 10^−213^) and of thiolases (e-value: 7.42 × 10^−167^), which participate in the butyrate synthesis pathways in bacteria [121], as well as genes related to 3-hydroxy-isobutyrate dehydrogenase, related to the metabolism of butyric acid (e-value: 3.26 × 10^−62^). With respect to *L. rhamnosus* UCO-25A, we used a *L. rhamnosus* strain, LRB, as a reference for this analysis, which shares a 99.93% identity with strain UCO-25A; it possesses genes related to the synthesis of hydroxyglutaryl (e-value: 1.11 × 10^−279^) and thiolases (e-value: 7.59 × 10^−268^) involved in butyrate synthesis. With regard to short-chain fatty acids, genes related to their transport (9.61 × 10^−49^), to condensation reactions which initiate the synthesis of these fatty acids (e-value: 5.19 × 10^−224^), and genes related to their biosynthesis (e-value: 3.5 × 10^−100^) are present.

On the issue of bacteriocins, the analysis of *L. fermentum* UCO-979C detected the presence of a gene from the family of linocin M18 (related to a bacteriocin) (e-value: 5.9 × 10^−148^) with 94.29% identity (e-value: 2 × 10^−180^). Linocin M18 favors the humoral immune response in the lungs and stimulates strong cytokine responses [122]. The analysis of *L. rhamnosus* UCO-25A revealed the presence of genes related to the secretion of bacteriocins (e-value: 1.73 × 10^−309^) and to bacteriocin proper (2.31 × 10^−120^).

Studies have shown that secondary bile acids have a significant impact on the regulation of the immune response. It has been observed that the bacterial metabolism of secondary bile acids is capable of stimulating the production of Treg cells [123] and of regulating inflammatory responses [124]. Therefore, the search for genes related to this metabolism becomes an interesting issue. The analyses found that *L. fermentum* UCO-979C possesses genes related to bile acid dehydratase (e-value: 4.55 × 10^−171^), related to the synthesis of secondary bile acids [125]. In the case of *L. rhamnosus* UCO-25A, genes related to the transport of bile acids (e-value: 1.59 × 10^−218^) were found.

**Table 3 biology-12-00384-t003:** In silico analysis of genes codifying the production of metabolites in *L. fermentum* UCO-979C and *L. rhamnosus* UCO-25A strains which could have an effect on infection by SARS-CoV-2 virus.

Probiotic Strain	Metabolite	Gene	Related Gene UniProt Code (Organism)	Function Related to Pathogenicity	References
*L. fermentum* UCO-979C	Short-chain fatty acid	3-hydroxy-isobutyrate dehydrogenase	P31937 (human)	Participates in the 3- hidroxybutyrate synthesis pathway in bacteria	[126]
		Thiolases	D7GV33 (butyrate-producing bacteria)	Participates in the butyrate synthesis pathway via acetyl-CoA	[121]
	Bacteriocins	Genes related to the linocin M18 family	V0A9Z1 (E. coli)	Bacteriocin that is capable of stimulating the humoral immune response in the lungs	[122]
	Secondary bile acids	Bile acid dehydratase	P19412 (Clostridium scindens)	Participates in the biosynthesis of bile acids, which stimulate the immune response associated with T cells	[125]
*L. rhamnosus* UCO-25A	Short-chain fatty acid	Genes related to the synthesis of hydroxyglutaryl	A0A2P6WFN2 (acidobacteria bacterium)	Participates in the butyrate synthesis pathway via glutarate	[121]
		Thiolases	D7GV33 (butyrate-producing bacteria)	Participates in the butyrate synthesis pathway via acetyl-CoA	[121]
		Genes related to their biosynthesis	P54616 (Bacillus subtilis)	Participates in the synthesis of short-chain fatty acids, which have immunomodulatory functions	[100,101,102]
	Bacteriocins	Bacteriocin	P83002 (Lactococcus lactis subsp. lactis)	May be able to prevent the entry of SARS-CoV-2 into host cells and exhibit antiviral activity	[91,122]
		Genes related to secretion	P22519 (*E. coli*)	Participates in the secretion of bacteriocins from the probiotic strains to exert their antiviral function	[91,122]
	Secondary bile acids	Genes related to transport	P32369 (Clostridium scindens)	Participates in the transport of secondary bile acids, which stimulate the immune system	[123,124]

## 4. Discussion

After gathering the precedents and information on SARS-CoV-2-caused infection, as well as the research performed at our Laboratory of Bacterial Pathogenicity at the University of Concepcion on the immunomodulating *L. fermentum* UCO-979C and *L. rhamnosus* UCO-25A strains, we can say that both strains may have a potential use against SARS-CoV-2 infection. This statement is based on the evaluation of the effects and symptomatology of SARS-CoV-2-caused infection and the probiotic as well as immunomodulating characteristics of both strains. The deregulation of several cytokines during the course of SARS-CoV-2 infection triggers a “cytokine storm”, causing a deregulation of different components of the immune system, generating lung hyperinflammation, which may lead to the death of a patient. Immunomodulating probiotic strains could have an effect on this deregulation through the intestine–lung axis, regulating the different cytokines and other components of the immune system, as stated in Section 3.8., possibly leading to the mitigation of lung hyperinflammation.

Moreover, SARS-CoV-2 virus infection can also cause gastrointestinal symptoms in nearly 10% of infected patients. These cases suffer intestinal dysbiosis, which is accompanied by the deregulation of cytokines and other components of the immune system. Confronting the symptomatology of intestinal SARS-CoV-2 infection, immunoregulatory probiotic strains could drive dysbiosis to a condition of eubiosis, reducing the gastrointestinal symptoms and regulating the proinflammatory as well as anti-inflammatory factors of the gastrointestinal immune system, as indicated in Section 3.8.

The bioinformatics analysis of *L. fermentum* UCO-979C and *L. rhamnosus* UCO-25A strains has allowed us, firstly, to observe the potentiality of both strains to be used as a possible complementary treatment against COVID-19. In addition to the immunomodulating effects already analyzed in this review, secondary metabolites have become highly relevant in the search for possible treatments against COVID-19 [89]. In this context, a bioinformatics search for genes associated with the regulation of the immune response and the inflammatory response was carried out. Several genes related to the synthesis and metabolism of butyrate, one of the short-chain fatty acids that have demonstrated several benefits in terms of the regulation of inflammation and immunomodulation [100], were found in *L. fermentum* UCO-979C. In addition, genes related to the synthesis of secondary bile acids, which participate in the regulation of the immune and inflammatory responses [124], were also found in *L. fermentum* UCO-979C. On the other hand, *L. rhamnosus* UCO-25A may have genes that participate in the synthesis of butyrate and genes related to the synthesis and transport of other types of short-chain fatty acids besides butyrate [127]. In the first approach, no genes related to the synthesis of secondary bile salts were observed in *L. rhamnosus* UCO-25A, but genes related to the transport of these metabolites were found. Thus, an effect of this probiotic by means of this pathway cannot be ruled out yet.

Finally, as previously mentioned, bacteriocins have shown a positive effect on the reduction in SARS-CoV-2 infection, as is the case of plantaricin [77]. The in silico analysis demonstrated the presence of genes related to bacteriocins in both strains. Lincocin M18, a stimulator of the cytokine’s response in the lungs [122], was detected in *L. fermentum* UCO-979C. This same type of analysis detected an unidentified type of bacteriocin in *L. rhamnosus* UCO-25A. Since previous studies of our laboratory have demonstrated the presence of a bacteriocin of the acidocin type in this strain, both results may be correlated.

This analysis provides a good approach to the metabolites of the *L. fermentum* UCO-979C and *L. rhamnosus* UCO-25A strains, which could have an effect on infection by SARS-CoV-2, allowing us to better focus future studies to develop an immunobiotic to be used as a complement to treat COVID-19.

## 5. Conclusions

This review addresses SARS-CoV-2 infection from the point of view of probiotics, mainly analyzing the symptomatology associated with infection and how taking advantage of the characteristics of probiotics as well as their metabolites could contribute to dealing with COVID-19.

As an example of probiotics, an in-depth analysis of two immunomodulating strains, *L. fermentum* UCO-979C and *L. rhamnosus* UCO-25A, of the Laboratory of Bacterial Pathogenicity, University of Concepción (Chile), have shown to possibly be beneficial when confronting infection by SAR-CoV-2. The search for commercial products and patents to treat COVID-19 opens the possibility of scaling the culturing of strains, improving the technology to generate a commercial product with a large health and social impact for the community.

This review also demonstrates the presence in the probiotic strains *L. fermentum* UCO-979C and *L. rhamnosus* UCO-25A of different genes of interest that could have anti-COVID-19 effects. An important issue is the search for a product capable of using the probiotic strains *L. fermentum* UCO-979C and *L. rhamnosus* UCO-25A with prophylactic purposes, an aspect not referred to by the analyzed patents; that is to say, to obtain an anti-SARS-CoV-2 effect different to those already existing.

## Figures and Tables

**Figure 1 biology-12-00384-f001:**
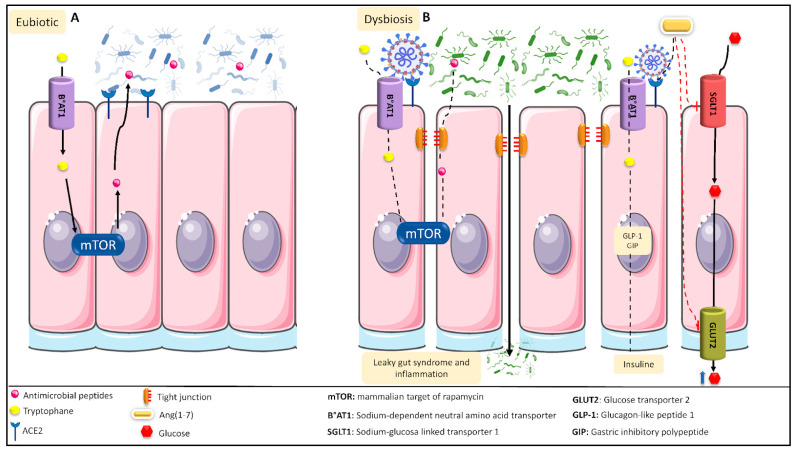
Intestinal eubiotic and SARS-CoV-2-induced intestinal dysbiosis. (**A**) In a general intestinal environment in eubiotic conditions, tryptophan can enter into the intestinal tract’s epithelial cells, generating the production of antimicrobial peptides that will regulate the intestinal microbiota. In addition, ACE2 can convert Ang II into Ang (1-7), which, after binding to its receptor, Mas, blocks intestinal glucose transport. (**B**) The various proinflammatory mechanisms caused during SARS-CoV-2 infection generate an intestinal dysbiosis condition. Under this condition, SARS-CoV-2 can block the B°AT1 receptor, preventing both the passage of tryptophan into the epithelial cells and the production of antimicrobial peptides. SARS-CoV-2 can also disrupt the tight junctions of the cells, producing leaky gut syndrome and inflammation. Moreover, SARS-CoV-2 binds to the ACE2 receptor, preventing the formation of Ang (1-7) and favoring the activity of the SGLT1 as well as GLUT2 transporters, increasing the intestinal glucose levels.

**Figure 2 biology-12-00384-f002:**
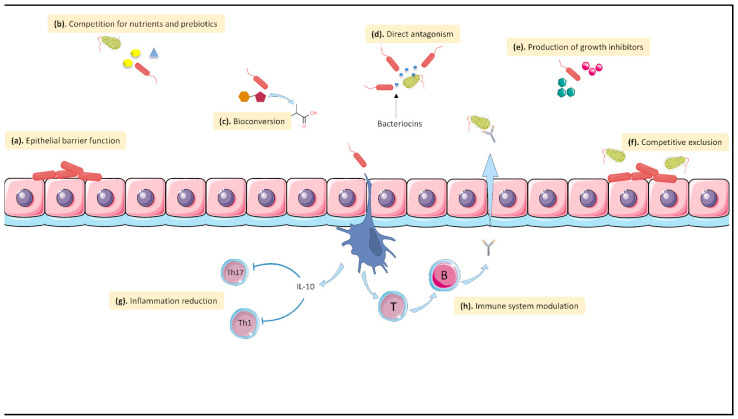
Mechanisms of action of probiotics. Several mechanisms are associated with the antagonistic effect of probiotics against intestinal pathogens, including the following: (**a**). Enhancement of the epithelial barrier function. Probiotic bacteria can maintain the barrier’s integrity, favoring its adhesion to the intestinal mucosa and maintaining the epithelial junctional complex, preventing the entry of pathogens or metabolites that may cause inflammatory responses. (**b**). Competition for nutrients and prebiotics. Probiotic bacteria are capable of competing for nutrients that are necessary for the survival of pathogens. (**c**). Bioconversion. Commensal bacteria can transform organic compounds into bioactive metabolites to generate a better environment. (**d**). Direct antagonism. Probiotic bacteria can produce bacteriocins and other compounds that can attack pathogens. (**e**). Production of growth inhibitors. Probiotic bacteria can produce compounds that inhibit the growth of some pathogenic bacteria. (**f**). Competitive exclusion. The mechanism by which certain bacteria are better competitors than pathogens for receptors. (**g**). Inflammation reduction. Commensal bacteria are capable of inducing an anti-inflammatory response, causing the inhibition of Th1, Th2, and Th17. (**h**). Immune system modulation. Bacteria can interact with dendritic cells, causing an immunomodulating effect and inducing B and T cell responses.

**Figure 3 biology-12-00384-f003:**
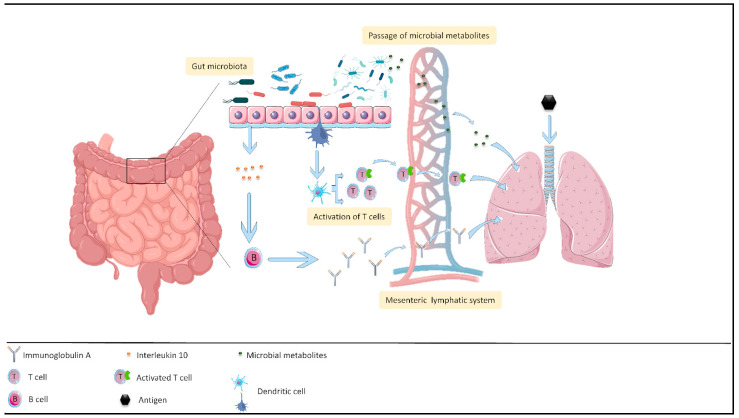
The intestine–lung axis. The regulation of lung immunity by the gastrointestinal microbiota is based on three basic alternatives. The first is the production of antibodies (immunoglobulin A) by activating B cells during the stimulation of the immune system by commensal intestinal bacteria. The second is the activation of T cells, stimulated by the dendritic cells interacting with commensal intestinal bacteria. The third is the generation of bacterial metabolites by the intestinal microbiota. All three can pass by the lymphatic system and reach the lungs, where they can generate responses against respiratory pathogens.

**Figure 4 biology-12-00384-f004:**
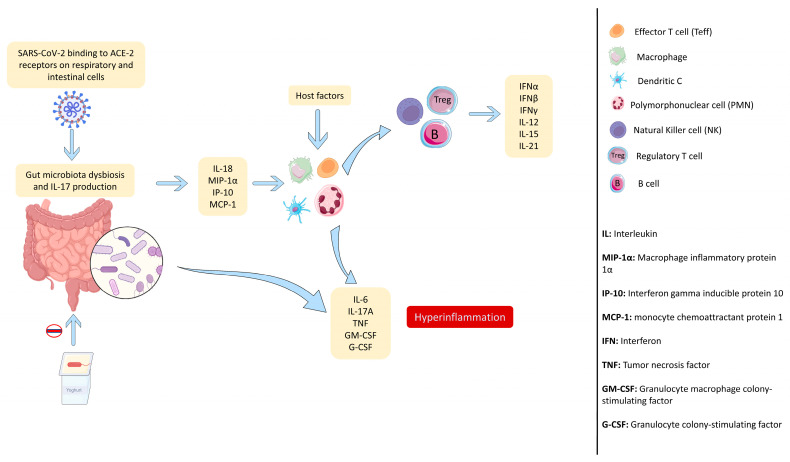
Mechanism of action of probiotics against SARS-CoV-2 infection. SARS-CoV-2 binds to ACE2 respiratory and intestinal cell receptors, producing dysbiosis of the intestinal microbiota. Probiotics can induce innate-immunity-activating cells, such as NK cells, polymorphonuclear (PMN) leukocytes, and macrophages, by means of the activity of cytokines, triggering responses with an immunomodulatory effect and also being capable of producing regulated inflammatory reactions through the action of cytokines.

**Table 1 biology-12-00384-t001:** Probiotics studied in humans that might contribute to decrease respiratory tract infection outbreaks.

Probiotic	Function of the Probiotic	Reference
*Lactobacillus casei* DN-114 001; fermented beverage DanActive/Actimel, Danone.	Reduces the incidence and duration of respiratory tract infections.	[62]
*Lactobacillus gasseri* PA 16/8, *Bifidobacterium longum* SP 07/3, and *B. bifidum* MF 20/5; TriBion Harmonis, Merck.	Reduces the duration and acuteness of influenza-type infections.	[63]
*Lactobacillus rhamnosus* GG; Culturelle or other brands.	For healthy digestion and a strong intestinal barrier. Anticipation of a respiratory tract viral infection.	[64]
*Lactobacillus plantarum* DR7.	Immunologic stimulation reduces infections of the superior respiratory tract.	[65]
*Bifidobacterium* breve Yakult and *Lactobacillus casei* Shirota; available as fermented beverages.	Reduces the prevalence of ventilator-associated pneumonia. The oral administration of *L. casei* Shirota to neonatal and infant mice showed that a boost of the immature immune system can play a positive role in the protection against influenza virus infection.	[4,50,66]
*Bifidobacterium longum* BB536; Morinaga, sold in many formulations.	Stimulates the innate immune response and protects against influenza.	[67]
*Lactobacillus rhamnosus* CRL1505.	Administered in yogurt, it improves mucosal immunity and reduces the incidence as well as severity of intestinal and respiratory viral infections.	[26]
*Lactobacillus acidophilus* NCFM.	Reduces the occurrence of influenza-like symptoms.	[4,68]
*Lactobacillus plantarum* YU, *Lactobacillus plantarum* L-137.	Stimulates the immune system of a host by lactic acid bacteria (LAB) against the influenza virus H1N1 and provides protective effects against some respiratory virus infections in mouse models and humans.	[4,69,70]
*Lactobacillus fermentum* CECT5716, *Lactobacillus casei* DN-114 001.	Probiotics have enhanced the effects of the influenza virus vaccine.	[4,71,72]
*Lactobacillus rhamnosus GG, Lactobacillus casei* (including the Shirota strain), *Lactobacillus plantarum, Bifidobacterium lactis* Bb-12, and various strains of *Bifidobacterium longum*.	Significant reduction in the prevalence of upper respiratory infections and flu-related symptoms.	[4,73]

## Data Availability

Not applicable.

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
