# Peer review of "Can Probiotics, Particularly *Limosilactobacillus fermentum* UCO-979C and *Lacticaseibacillus rhamnosus* UCO-25A, Be Preventive Alternatives against SARS-CoV-2?"

_biology, 2023, doi:10.3390/biology12030384_

Round 1

Reviewer 1 Report

This manuscript has fallen into several pitfalls; most importantly, the type of the article is not clear. Although it is presented as a literature review, the authors also present data.

The aims and objectives as presented in the abstract as rather confusing and unclear.

The title is not representative of the article.

Most of the data presented in the introduction are well-known facts described multiple times in the literature, this info could be condensed in a new Introduction section. I would also suggest to the authors focus more on this section on the potential effects of probiotics against viral disease, as well as on the intestinal effects of COVID-19, which could be reversed with probiotic supplementation.

In Lines 193-199 the authors present how they searched the literature for articles, why did the authors search for articles published from 2015 to 2022 for these keywords, when SARS-CoV-2 was first described in 2020?

Table 2: references should be added.

Lines 495-498: predictions are very important to steer research towards new avenues, however, a comment should be added about in vitro, in vivo validation of these in silico findings.

Ι would suggest that the authors focus more on the bioinformatics data, performing more extensive comparative analyses and genomics approaches to determine the possible functionality of the metabolites identified, using the literature review to support their findings.

Author Response

Answers in the uploaded pdf file.

Reviewer 2 Report

biology-2108462-peer-review-v1

The present paper provides overview on covid-19 infection, history and trends in development of vaccines and potential clinical treatments and prevention strategies. In my opinion paper is interesting and deserve to be considered for publication, however, some adjustments need to be taken into consideration by authors.

Ln177: Maybe will be good if information for the company develops these drugs will be provided.

Ln181-183: Mentioned approaches are more complementary treatment then principal drugs. Maybe this need to be mentioned.

Ln277: remove one "." at the end of the sentence.

Please, for the systematic names of mentioned bacterial strains, use recommendations from April 2020.

Some probiotics are suggested to be applied as live vaccines in control of influenca and other respiratory infections. Maybe information about them can be added to the Table 1? Please, check doi: 10.1007/s12602-021-09833-0.

Table 1: remove italics form CRL1505

Ln286-293: Maybe will be good if can be mentioned numbers of individuals participating in this clinical trial?

Ln314: Maybe in this place and following examples, this can be stated as "3.5.1.1. RBD", then on Ln325: "3.5.1.2. 3CLpro", etc..

Ln315: coronavirus or coronaviruses?

Ln322: Please, provide full name of the mentioned plantaricin and strain identification for Lactiplantibacillus plantarum, strain producing mentioned bacteriocin.

Ln324: Please, correct to "(RdRp) [65]."

Ln326-333: Any proteinase inhibitor that can target that 3CLpro? You have mentioned on Ln331-333 that some compounds can interfere with 3CLpro, please, add a few sentences more on this topic and provide appropriate references.

Ln346: Please, consider to correct to "Sui et al. [72]"

Ln361: Are only plantaricin responsible for such as effect? Some studies have shown that other bacteriocins (produced by different lactic acid bacteria) can have some antiviral effects. Maybe in addition need to be provided a bit more information regarding the topic.

Maybe this subtitle can be changes to "bacteriocins from lactic acid bacteria" and then provide information about mentioned plantaricin and add additional information about other bacteriocins.

Ln362: Most probably mentioned properties are strain specific and do not species specific characteristics. Thus, will be appropriate to mention strain identifications for mentioned Lactiplantibacillus plantarum strain.

Ln363: Please, consider to correct to "Anwar et al. [65]". Please, check and correct rest of the manuscript for similar adjustments.

Ln380: correct to "[80-82]"

Ln388-391: This information can be move with a previous topic and be enriched a bit more.

Ln491-492: Provided information in Table 2 was based on research performed for this manuscript, or based on previously performed and published studies? Maybe in title of the table will be appropriate to say "Based on previous studies" and mention them as references.

Ln499: Please, mention that this is done based on metagenomic analysis in silico.

Author Response

Answers in the uploaded pdf file.

Round 2

Reviewer 1 Report

The authors have satisfactorily addressed most of my comments. The resubmitted manuscript is suitable for publication.